# Precise control of embolic stroke with magnetized red blood cells in mice

Yuxiao Jin [1,2,3,4], Peijun Shi[5,6], Yu Wang[7], Jinghang Li [6,8], Jiachen Zhang[9], Xinxin Zhao[10], Yaping Ge[2,3,4], Yanjie Huang[2], Mengzhun Guo[7,11], Feidi Wang[12], Bo Ci[13], Xian Xiao[2,3], Xiaofei Gao[2,3], Jianrong Xu[10], Bobo Dang [7,11], Botao Ji [5,6], Woo-ping Ge [14] & Jie-Min Jia [2,3,4 ✉]

Precise embolism control in immature brains can facilitate mechanistic studies of brain damage and repair after perinatal arterial ischemic stroke (PAIS), but it remains a technical challenge. Microhemorrhagic transformation is observed in one-third of infant patients who have suffered PAIS, but the underlying mechanism remains elusive. Building on an established approach that uses magnetic nanoparticles to induce PAIS, we develop a more advanced approach that utilizes magnetized erythrocytes to precisely manipulate de novo and in situ embolus formation and reperfusion in perinatal rodent brains. This approach grants spatiotemporal control of embolic stroke without any transarterial delivery of pre-formed emboli. Transmission electron microscopy revealed that erythrocytes rather than nanoparticles are the main material obstructing the vessels. Both approaches can induce microbleeds as an age-dependent complication; this complication can be prevented by microglia and macrophage depletion. Thus, this study provides an animal model mimicking perinatal embolic stroke and implies a potential therapeutic strategy for the treatment of perinatal stroke.

[1] College of Life Sciences, Zhejiang University, Hangzhou, Zhejiang 310058, China. [2] Key Laboratory of Growth Regulation and Translational Research of Zhejiang Province, School of Life Sciences, Westlake University, Hangzhou 310024, China. [3] Westlake Laboratory of Life Sciences and Biomedicine, Hangzhou 310024, China. [4] Laboratory of Neurovascular Biology, Institute of Basic Medical Sciences, Westlake Institute for Advanced Study, Hangzhou 310024, China. [5] Key Laboratory of 3D Micro/Nano Fabrication and Characterization of Zhejiang Province, Hangzhou, China. [6] School of Engineering, Westlake University and Institute of Advanced Technology, Westlake Institute for Advanced Study, Hangzhou, China. [7] Westlake Laboratory of Life Sciences and Biomedicine, Key Laboratory of Structural Biology of Zhejiang Province, School of Life Sciences, Westlake University, Hangzhou, Zhejiang, China. [8] School of Materials Science and Engineering, Wuhan Institute of Technology, Wuhan, Hubei, China. [9] School of Basic Medical Sciences, Wuhan University, Wuhan, China. [10] Department of Radiology, Ren Ji Hospital, School of Medicine, Shanghai Jiao Tong University, Shanghai, China. [11] Institute of Biology, Westlake Institute for Advanced Study, Hangzhou, Zhejiang, China. [12] Department of Anesthesiology & Center for Brain Science, The First Affiliated Hospital of Xi'an Jiaotong University, Xi'an, Shaanxi Province, China. [13] Children's Medical Center Research Institute, University of Texas Southwestern Medical Center, Dallas, TX, USA. [14] Chinese Institute for Brain Research, Beijing, China. ✉email: jiajiemin@westlake.edu.cn

Perinatal stroke is a serious neurological disease that affects fetuses, preterm infants, and full-term infants; it is defined as cerebrovascular injury occurring between 20 weeks of pregnancy and 28 days after birth[1,2]. A recent report revealed that the birth prevalence of perinatal stroke (1:1100) in full-term infants was higher than that in any of the previous reports and comparable to the incidence in the elderly population[3]. The estimated risk for premature infants is believed to be 100 times higher than that of full-term infants[4]. Perinatal arterial ischemic stroke (PAIS) is the most common type among the diverse subtypes of perinatal stroke[3,5]. PAIS is clinically mostly asymptomatic, so its incidence has been underestimated[6], and its symptoms generally emerge after the neonatal period, such as cerebral palsy and cognitive dysfunction, which is also one of the reasons why PAIS seriously affects long-term disability and teratogenesis in neonates. Approximately 90% of PAIS cases occur in the middle cerebral artery (MCA)[7]. Unlike adult ischemic stroke, which is mainly caused by the progressive development of 'white' thrombosis and the resultant artery stenosis, PAIS is dominantly caused by a 'red' embolus, which is mainly made up of red blood cells and fibrins and distally originates from the placenta or the malfunctioning heart[2]. To elucidate the fundamental molecular and cellular mechanism of brain damage and repair after embolic stroke, diverse approaches have been produced, such as arterial delivery of pre-formed macro- or microemboli[8,9], local thrombin injection[10], rose bengal–based photothrombosis or ultrashort laser pulse–induced thrombosis at the MCA[11]. Although these models are closer to clinical stroke than mechanical occlusion approaches[12–14] and are well documented in adult animals, they are not feasible in mice at the age of P0-7 because the arterial diameter is much smaller than that of adult mice. In addition, these traditional methods neither avoid arterial surgery nor produce reversible embolic occlusion in perinatal mice[15,16].

Hemorrhagic transformation can occur spontaneously in the infarct tissue, complicate ischemia, and is associated with worse outcomes. Limited epidemiological studies have shown that pediatric patients have five times the incidence of hemorrhagic transformation than adult patients after focal ischemic stroke[17–19]. During the subacute period, which is referred to as the time window from 1 to 5 days after ischemia onset, 30% of neonate patients have microbleeds as a complication[19–21], whereas less than 10% of adults do[22]. There have been few reports comparing the hemorrhagic transformation frequency at the acute stage between neonates and adults. This merits further investigation in both epidemiological and basic research fields.

Microglia, the resident immune cells of the brain, are associated with cerebral vasculature and are among the first cells that respond to brain injury. The number of perivascular microglia increases gradually with stroke duration and activated microglia can phagocytose endothelial cells and promote blood vessel disintegration[23]. In addition, blood-derived macrophages penetrate the blood-brain barrier after attaching to endothelial cells and infiltrate into the brain parenchyma, exacerbate inflammation and tissue damage, and break down the integrity of the blood-brain barrier[24,25]. It has been reported that hemorrhagic incidence during the subacute period is decreased in adult mice by using minocycline, which inhibits microglia and macrophages[24]. However, the cellular mechanism underlying hemorrhagic transformation after perinatal arterial ischemic stroke has yet to be determined.

Here, we report the development of an approach named embolic stroke induced by magnetic nanoparticle-coated red blood cells (SIMPLeR). We were able to generate red thrombi *de novo* and in situ by aggregating magnetized red blood cells (mRBCs) at the distal MCA (dMCA) of mouse pups at the age of P0-7. We found that at the acute stage, perinatal mice had nine times the frequency of microhemorrhagic transformation as adults. The SIMPLeR model is an advanced version of the SIMPLE model (stroke induced by magnetic nanoparticles)[26] in the scenario of perinatal arterial embolic stroke in which most cases are caused by distally formed RBC-rich clots. Depletion of microglia and macrophages prevented microbleed complications.

## Results

**Production of magnetized red blood cells (mRBCs).** To produce mRBCs by connecting magnetic nanoparticles (MNPs) to red blood cells, we utilized two distinct methods (Fig. 1a and Supplementary Fig. 1a). First, we used the principle of streptavidin-biotin interaction to attach MNPs to the RBC membrane (Fig. 1a and a-i). We linked biotin to the RBC membrane by using an anti-Ter119 antibody conjugated with biotin and specifically recognized the RBC membrane protein marker Ter119. Then, the resultant biotin-coated RBCs were incubated with streptavidin-modified MNPs (streptavidin-MNPs; see Fig. 1a-ii–a-iii and Methods). Thus, mRBCs were successfully produced via a strong interaction between streptavidin and biotin (Fig. 1b–f). Compared to the smooth surface of unmagnetized RBCs (Fig. 1b, d), scanning and transmission electron microscopy revealed that we successfully adhered MNPs (magenta arrows) to RBCs in vitro (Fig. 1c, e). MNPs did not detach from mRBCs after administration into the bloodstream (Fig. 1f, magenta arrows). We used these electron microscopic images to quantify the ratio between the free nanoparticles that did not bind to RBCs versus the total nanoparticles we used. This quantification revealed that most MNPs (92%) were tagged onto RBCs, and only 8% of them were free. Thus, the first method based on streptavidin-biotin binding can, with a high efficiency of 92%, tag MNPs onto the RBC membrane. In addition, the RBC quality was closely monitored by microscopic inspection at every single step (Supplementary Fig. 2). We ceased the mRBC preparation process if we found that under 90% of erythrocytes had a normal biconcave shape.

Alternatively, we aimed to load MNPs into RBC cytoplasm by using a classical DNA transfection reagent—Lipofectamine (Supplementary Fig. 1a). However, Lipofectamine failed to do so; instead, it tethered MNPs to the cell membranes of RBCs, as revealed by electron microscopy (Supplementary Fig. 1b–f). Compared to the smooth surface of unmagnetized RBCs (Supplementary Fig. 1b-b-i and 1e), MNPs were attached onto RBCs (yellow arrows, Supplementary Fig. 1c). High-magnification images show MNPs partially embedded into the dint of the RBC membrane (magenta arrow, Supplementary Fig. 1c–i). Some RBCs seemed structurally damaged (green arrows in Supplementary Fig. 1d), and the membrane of RBCs had holes (blue arrows in Supplementary Fig. 1d–i), implying that this method may harm RBCs. These results demonstrated that although mRBC was successfully produced as well, the second method based on Lipofectamine might have poorer quality than the first method. Both strategies were carried out throughout the entire study, unless otherwise indicated.

To assess how these mRBCs respond to the magnetic field gradient, we produced mRBCs using the two methods mentioned above and performed an mRBC enrichment assay that was achieved by providing a magnetic field to a drop of liquid that may contain MNPs alone, mRBCs or unmagnetized RBCs (see Methods). MNPs were rapidly pooled together and aggregated into a line (black arrows, Fig. 1g; Movie 1), as were mRBCs (green arrows, Fig. 1g–i; Movie 2). In contrast, unmagnetized RBCs remained steady state (Fig. 1g-ii; movie 3). The aggregation speed of mRBCs was slower than that of MNPs alone. mRBC took 90 s to accumulate into a line, whereas MNP took only 30 s (Fig. 1g

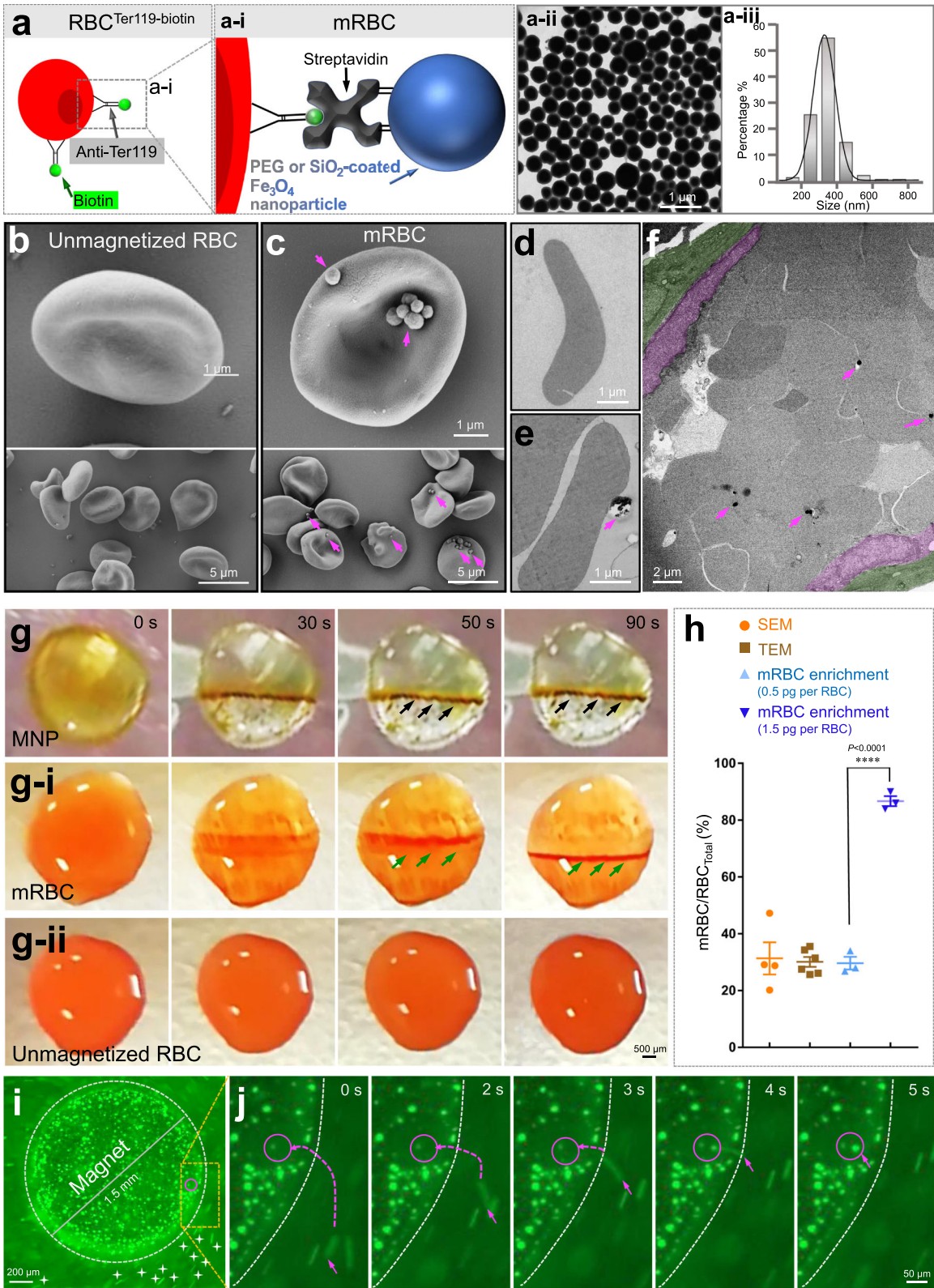

vs 1g-i). This result suggested that the weight and volume of RBCs may contribute to the slower motion of mRBCs.

To determine the RBC magnetization efficiency, we used the mRBC enrichment assay to calculate the percentage of unmagnetized RBCs. At an MNP dose of 0.5 pg per RBC for the magnetization reaction, we found that ~70% of RBCs were not magnetic. In other words, >30% of RBCs were magnetized

(Fig. 1h, light blue dots). This result was consistent with the results from electron microscopy imaging by counting mRBCs versus total RBCs (scanning electron microscopy, 31.3%, 148 of 427 cells, transmission electron microscopy, 30.1%, 196 of 654 cells, Fig. 1h, dark and light brown dots). These results indicated that the mRBC enrichment assay can indeed reflect the magnetization efficiency of RBCs. Therefore, we further assessed

**Fig. 1 Produce magnetized RBC (mRBC) in vitro. a-a-i** Schematic illustration of the principle of streptavidin-biotin-based mRBC generation. **a-ii** TEM image and particle diameter analysis **a-iii** of streptavidin-MNPs synthesized by our collaborators. Representative SEM (**b**, **c**) and TEM (**d**, **e**) images of unmagnetized RBCs (**b**, **d**) and mRBCs (**c**, **e**), respectively. Magenta arrows indicate MNPs. **f** TEM image of ultrathin sections from a segment of the middle cerebral artery, in which mRBCs accumulated. Endothelia and vascular smooth muscle cells are highlighted in magenta and green. Magenta arrows indicate MNPs. Time-lapse pictures of MNP (**g**), mRBC (**g-i**) and unmagnetized RBC (**g-ii**) enrichment upon magnetic gradient. Black arrows indicate aggregated MNPs, and green arrows show accumulated mRBCs. **h** Quantitative analysis of the RBC magnetization efficiency (SEM, 31.3%, 148 of 427 cells; TEM, 30.1%, 196 of 654 cells, three independent experiments were performed for each group of 'mRBC enrichment'). The effect size between two groups at doses of 0.5 and 1.5 pg per RBC is 18.2. **i** Snapshot of trapped DiO-mRBCs on a cylindrical magnet with a 1.5-mm diameter. The dashed circle indicates the edge of the magnet, and white stars indicate the floating unmagnetized DiO-RBCs and DiO-mRBCs. **j** Still images of the trajectory of one mRBC movement indicated with dashed magenta arrow.

the dose-dependency of RBC magnetization. We found that mRBC yield was augmented when MNP dosage gradually increased from 0.2 to 1.5 pg per RBC and turned to a decline at the higher dosages (Supplementary Fig. 1g). The highest dose of 13.5 pg per RBC killed RBCs directly (Supplementary Fig. 1g). Notably, at the 1.5 pg per RBC reaction dosage, we found that only 16% of RBCs were not magnetic, indicating that more than 80% of total RBCs were magnetized (Fig. 1h, darker blue dots; Supplementary Fig. 1g). Hereafter, a dosage of 1.5 pg per RBC was used throughout the following study. These results together indicated that the RBC magnetization efficiency was dose dependent and that an overdose of MNPs (13.5 pg per RBC) damaged RBCs.

We video-recorded the entire process of an mRBC floating toward a magnet (Fig. 1i; Movie 4). The mRBCs used in this experiment were produced by the Lipofectamine-based method. All RBCs were stained green by using the lipophilic dye DiO. Immediately after adding the mixture of unmagnetized RBCs and mRBCs that were both DiO positive into a chamber with a cylindrical magnet (1.5-mm diameter) glued at the center, all the RBCs initially floated in stochastic directions in the unsettled buffer. However, 10 s later, we observed that many DiO-mRBCs were trapped onto the surface of the magnet (Fig. 1i, green dots), and other RBCs were still floating around (Fig. 1i, white stars, Movie 4). We particularly focused on the region indicated by the magenta circle inside the yellow boxed area. The floating path of one DiO-mRBC is traced by a long-curved magenta dashed arrow (Fig. 1j). We noticed that it took only 5 s for this DiO-mRBC to be attracted onto the magnet (magenta arrow, Fig. 1j, Movie 4). The possibility that gravity caused the RBCs to settle was ruled out because the erythrocyte sedimentation rate is on the time scale of minutes to hours, which is far behind the mRBC's motion. Taken together, these results demonstrated that mRBCs were successfully produced and responded to the magnetic field gradient rapidly and effectively in vitro.

**mRBCs aggregated in vivo in response to magnetic force**. Next, we aimed to aggregate the mRBCs in vivo (Fig. 2a). We first examined whether DiO-mRBCs (Lipofectamine-based method) remained magnetic after their entry into the adult mouse circulation. DiO-mRBCs that were drawn back from the bloodstream were aggregated by a magnetic field again, whereas unmagnetized DiO-RBCs were not (Fig. 2b, c). This result demonstrated that MNPs did not entirely dissociate from RBCs after circulating in the endogenous bloodstream environment.

Therefore, we directly monitored DiO-mRBC behavior in vivo by taking advantage of transparency of the skull of perinatal mouse pups (postnatal day 3, P3). DiO-mRBCs were delivered through the superficial temporal vein (Fig. 2a). Hereafter, all mRBCs were generated by using the first method based on the principle of streptavidin-biotin interaction. The injected mRBCs less than 1 million per gram of mouse body weight failed to aggregate in vivo (Table 1). Injection of 1-2 million DiO-

mRBCs g$^{-1}$ gradually aggregated in the microvessels around and under the magnet (Fig. 2d, e, magenta arrows and Table 1). It took only 3 s to form two additional DiO-mRBC clots (② vs. ③ in Fig. 2d). Twelve seconds later, these 3 initially separated clots were aggregated together (⑤ in Fig. 2d). These data demonstrated that a magnetic field was able to trap the flowing DiO-mRBCs at targeted vessels in a few seconds.

Alternatively, the magnet was kept on top of the vein sinus for 9 s and then removed. We found that DiO-mRBCs were trapped in the venous sinus, which has a large luminal space (③ vs. ① in Fig. 2e, magenta arrow). It took more than 30 s for those previously aggregated DiO-mRBCs to undergo gradual dispersal (③–⑤ in Fig. 2e, magenta arrow), indicating that we can manipulate mRBC motions. Apparently, these accumulated mRBCs did not fully obstruct the venous sinus. Thus, we increased the mRBC number up to 12 million g$^{-1}$ and resulted in full occlusion (Movie 5). In addition, we used Alexa Fluor 488-labeled mRBCs (Alexa Fluor 488-mRBCs) instead of DiO-mRBC. We intended to aggregate Alexa Fluor 488-mRBCs to induce a blood clot in the dMCA, which is the most affected vessel in ischemic stroke[27]. Notably, Alexa Fluor 488-mRBCs were selectively trapped at the targeted artery (magenta arrows in Fig. 2f, g), whereas barely any Alexa Fluor 488 signals were detected in the neighboring vessels (green arrows in Fig. 2f, g). This result demonstrated, again, that we were able to magnetize RBCs with high efficiency in vitro; otherwise, we would have observed unmagnetized Alex Fluor 488-RBCs in neighboring regions. The occlusion caused by the red thrombus in the distal middle cerebral artery of a P3 mouse was reversible (Movie 6). These experimental results strongly demonstrated that we can spatiotemporally control red thrombus formation and reperfusion at the distal middle cerebral artery of perinatal mouse pups by manipulating mRBCs.

**Perinatal arterial embolic stroke was induced by occluding the dMCA using SIMPLeR**. Given that 90% of PAIS cases occur at the MCA[7], we next aimed to test the capability of SIMPLeR to induce hypoperfusion in the territory of the dMCA. Assessed by laser speckle contrast imaging, SIMPLeR, using 6 mg kg$^{-1}$ MNPs, successfully achieved a regional reduction in blood supply by more than 50% in the brain region outlined by the magenta dashed line (Fig. 3a). Notably, SIMPLE, using the same low dose, failed (Fig. 3b). Only when the MNP dosage increased to 80 mg kg$^{-1}$ did SIMPLE achieve a similar degree of hypoperfusion as SIMPLeR (Fig. 3c, d). It took 30 min for the blood flow to reach the low perfusion plateau that lasted up to 10 h (Fig. 3d). This indicates that both SIMPLeR and SIMPLE could result in permanent occlusion if magnets were not removed (Fig. 3d). Furthermore, we used two-photon live imaging and revealed that the blood flow direction, as in adult mice[28,29], was disrupted in the arteriolar branches of the MCA following occlusion (Movie 7). The deposition of MNPs in SIMPLeR mouse livers, kidneys and spleens was negligible compared to that in SIMPLE

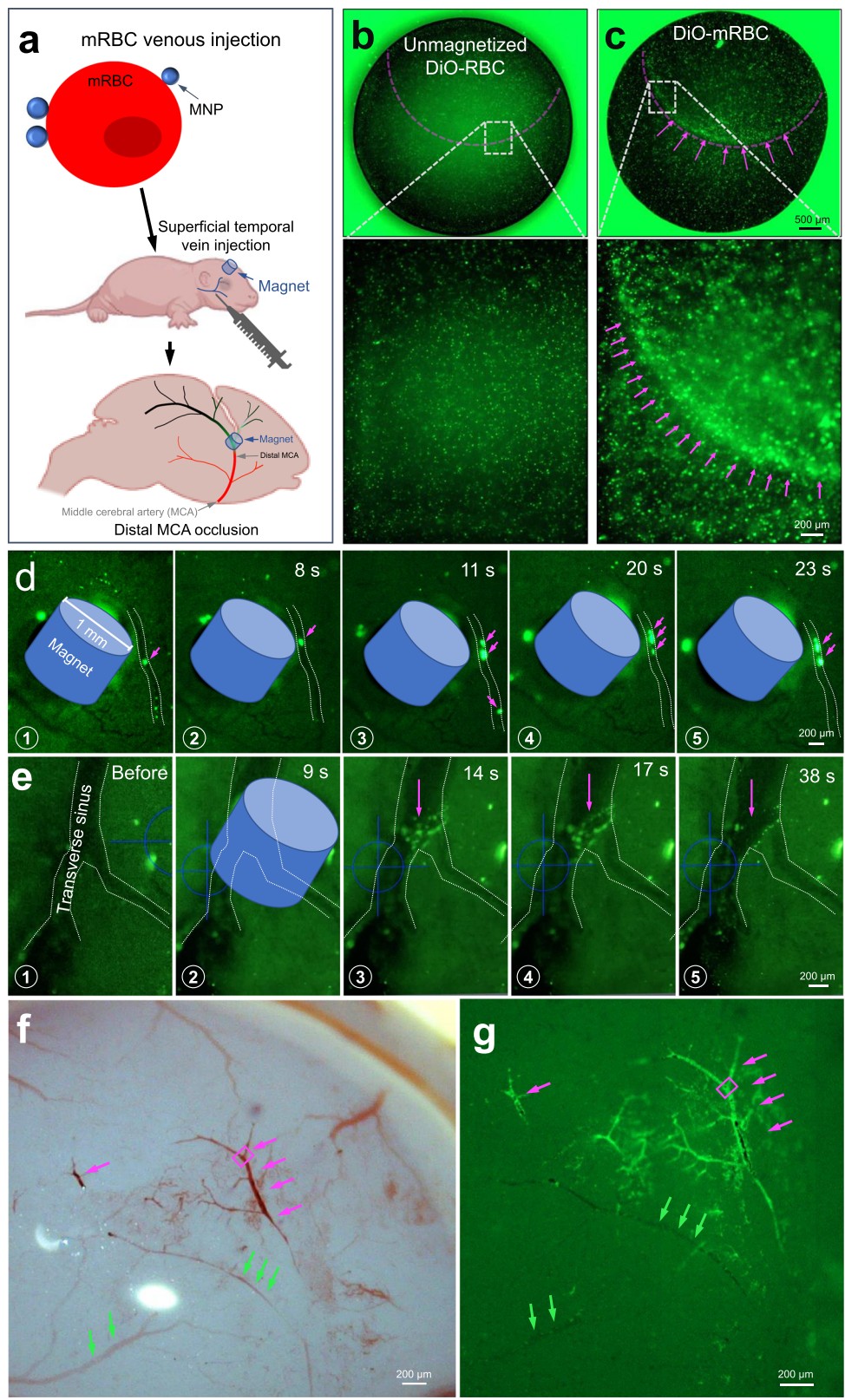

mice (Supplementary Fig. 3 and Movies 8–11). Thus, SIMPLeR is superior to SIMPLE by using a much lower amount of MNPs.

Extensive hypoperfusion leads to neuronal death. At 6 h after dMCA occlusion, SIMPLeR (6 mg kg⁻¹) induced massive neurodegeneration, as revealed by Fluoro-Jade C staining (Fig. 3e). SIMPLE did not cause any neuronal death when administered at 6 mg kg⁻¹ (Fig. 3f), but it did when administered at 80 mg kg⁻¹ (Fig. 3g-h).

The infarction volume was far beyond the aggregation site of mRBCs, as evidenced by the sections in Fig. 3e-g being at least 2 mm remotely posterior to the occlusion site in the dMCA. Neurodegeneration took place in the somatosensory cortex region with a sharp boundary separating the nonischemic area (Fig. 3e and g). Hoechst, NeuN, and GFAP staining revealed condensed nuclei, neuronal loss, and damaged radial glial processes in the ischemic area

**Fig. 2 mRBCs occlude the targeted microvessels in vivo. a** Schematic illustration of the principle of SIMPLeR by using a magnet to trap mRBCs at the distal MCA. Low- and high-magnification images of unmagnetized DiO-RBCs (**b**) and DiO-mRBCs (**c**) upon magnetic gradient. Both unmagnetized DiO-RBCs and DiO-mRBCs were withdrawn 5 min after they were injected into two adult mice. The magenta dashed line indicates the edge of a cylindrical magnet underneath. Magnet arrows indicate the accumulated mRBCs. Three independent experimental repeats were conducted. **d** Time-lapse fluorescent stereoscopic images of P3 mouse live brains, in which the process of gradual accumulation of DiO-mRBCs (magenta arrows) around the micromagnet (1 mm diameter) was recorded. **e** Still images of the dispersal process of mRBCs after the magnet was removed. Magenta arrows indicate the dispersing mRBCs. **f, g** Bright field (**f**) and fluorescent stereoscopic images of a P3 mouse brain, of which dMCA was occluded by Alexa Fluor 488-mRBCs (magenta arrows). Green arrows indicate the untargeted neighboring vessels. The segment of dMCA in the magenta box was subjected to TEM, as shown in Fig. 3n–o and Supplementary Fig. 5. The illustrations of mouse pup body and brain in (**a**) were copied from Biorender.com.

| Table 1 Relationship between the injected amount of mRBCs and vascular occlusion formation. | | | |
| --- | --- | --- | --- |
| Age | Weight (g) | mRBCs (million g$^{-1}$) | Occlusion formation |
| P8 | 3.23 | 2.14 | Yes |
| P8 | 3.32 | 1.53 | Yes |
| P8 | 3.12 | 1.41 | Yes |
| P8 | 3.07 | 1.33 | Yes |
| P8 | 3.44 | 0.89 | No |
| P8 | 3.55 | 0.86 | No |
| P8 | 3.36 | 0.61 | No |

(Supplementary Fig. 4a–c). Neuronal density decreased by 25% to 33% in both the SIMPLeR and SIMPLE models (Supplementary Fig. 4d). Low-dose SIMPLE (6 mg kg$^{-1}$) did not cause any detectable defects (Supplementary Fig. 4b, d). Approximately 15% of the whole brain volume of the P5 mouse pup was infarcted, as indicated by TTC staining (Fig. 3i), while T2-weighted MRI scanning revealed a smaller infarction size, 0.85% following a 7-hour occlusion (Fig. 3j and l) and 3.18% following a 14.5-hour occlusion (Fig. 3k and l). A second MRI scan of the same mouse from Fig. 3j in young adulthood (P50) revealed that the infarction evolved to a thinner somatosensory cortex (yellow arrows) compared to the healthy contralateral side (white arrows, Fig. 3m). The success rate of red thrombus formation in the dMCA induced by SIMPLeR was 100%, but the success rate of stroke induction was 71% because it was not necessary for all occlusions resulting in ischemia. The survival rate was 100%, as all the pups that were subjected to the SIMPLeR model survived (Supplementary Fig. 4e). Taken together, our experiments show that we successfully modeled perinatal arterial embolic stroke using the approach of aggregating mRBCs (SIMPLeR), which is superior to SIMPLE with several aspects, including using less MNP and closer to clinical embolism in fetuses, preterm infants, and full-term infants.

**SIMPLeR and SIMPLE utilized distinct materials to obstruct vessels.** Although the etiology of PAIS is not yet thoroughly clear, emerging evidence suggests that emboli generated in the placenta or heart atrium are most likely the major causes[2]. Moreover, there is almost no atherosclerosis that is often related to white thrombi in such early developing brains. It would be important to take a close look at the nature of the obstructions generated by these two models. At 30 min after SIMPLeR, transmission electron microscopy revealed that many RBCs, including both mRBCs and unmagnetized RBCs, were stuck at the dMCA segment (Fig. 3n and Supplementary Fig. 5), which was from the same segment boxed in Fig. 2f-g (see Supplementary Fig. 5). The large volume of RBCs made the major contribution (92.6%) to obstructing the vascular lumen, while the MNPs accounted for very little volume (1.75%, Fig. 3n, s). This shows the underlying reason why SIMPLeR that utilizes only a small amount of MNPs can cause cerebral infarction, which usually requires a higher dose of SIMPLE.

Intriguingly, at 6 h after SIMPLeR, the occlusive material evolved, indicated by the fact that fibrin emerged to enwrap RBCs (Fig. 3o and s-i). The component ratio of fibrin-like structures accounted for up to 39%, RBCs occupied 23% and 38% of the total area left, including platelets and white blood cells. (Fig. 3s-i). This red embolus-like structure mimics the emboli generated in the human heart atrium and deep vein thrombus. In contrast, SIMPLE used MNPs as the central component of the obstruction (green outline Fig. 3p and s-ii panel, Supplementary Fig. 6). However, interestingly, platelet aggregates and white cell accumulation were located at regions adjacent to MNP aggregates (Fig. 3q–r and Supplementary Fig. 6). These results suggested that SIMPLeR more closely simulated clinical emboli than SIMPLE.

**Depletion of CSF1R$^+$ macrophages prevented cerebral ischemia-induced microbleeds.** Hemorrhagic transformation occurs spontaneously in the infarcted tissue, complicates PAIS and is associated with worse outcomes[20,21]. However, the underlying cellular mechanisms are largely unknown[21,24,30]. Consistent with our previous report[31], both SIMPLeR (6 mg kg$^{-1}$) and SIMPLE (80 mg kg$^{-1}$) resulted in petechial microbleeds in P3 mouse pups (Fig. 4a, c), but a low dose of SIMPLE (6 mg kg$^{-1}$) did not (Fig. 4b). Notably, this study recorded the dynamic process of bleeding by using two-photon cranial imaging of mouse pups that underwent focal cerebral ischemia (Movie 12). These petechial hemorrhages were usually remote from the infarcted area with a rim of restricted diffusion (Fig. 4a and c), similar to clinical observations[32]. When subjected to *in utero* photothrombosis (see Methods), gestational day 19 mouse embryos exhibited micro-hemorrhage as well (Fig. 4d). However, it was not certain whether this hemorrhage resulted directly from endothelial damage or indirectly from photothrombosis-induced ischemia. Nonetheless, our model successfully mimicked microhemorrhagic transformation following PAIS.

We further ascertained the relationship between 'hemorrhagic transformation probability' and 'occlusion duration' by using P3 to P5 mouse pups. We found that 11 of 17 pups did not have hemorrhagic transformation at 2 h after ischemia (Fig. 4e left panel), and 6 of 17 pups (35%) had a very mild hemorrhagic transformation, observed as punctate microbleeds (white arrows in Fig. 4e-2h). The hemorrhagic transformation rate gradually increased and reached 92% pups (38 out of 41) when dMCA occlusion lasted for 7–10 h. Not only the hemorrhagic transformation rate but also the severity of microbleeds became more obvious (Fig. 4e, f). A European cooperative acute stroke study subdivided hemorrhagic transformation into the hemorrhagic infarction (HI) type (with punctate foci of petechial hemorrhage) and the parenchymal hematoma (PH) type (with more confluent foci)[20]. The microbleed pattern shifted from a micro-scale punctate shape (Fig. 4e-2h) to a petechial shape (gray arrows in Fig. 4e-h) and progressed further to confluent bleeding foci (black arrow in Fig. 4e-6h). A short ischemia duration (2–5 h) caused only minor hemorrhagic infarction (HI)-type transformation, whereas parenchymal hematoma (PH)-type transformation

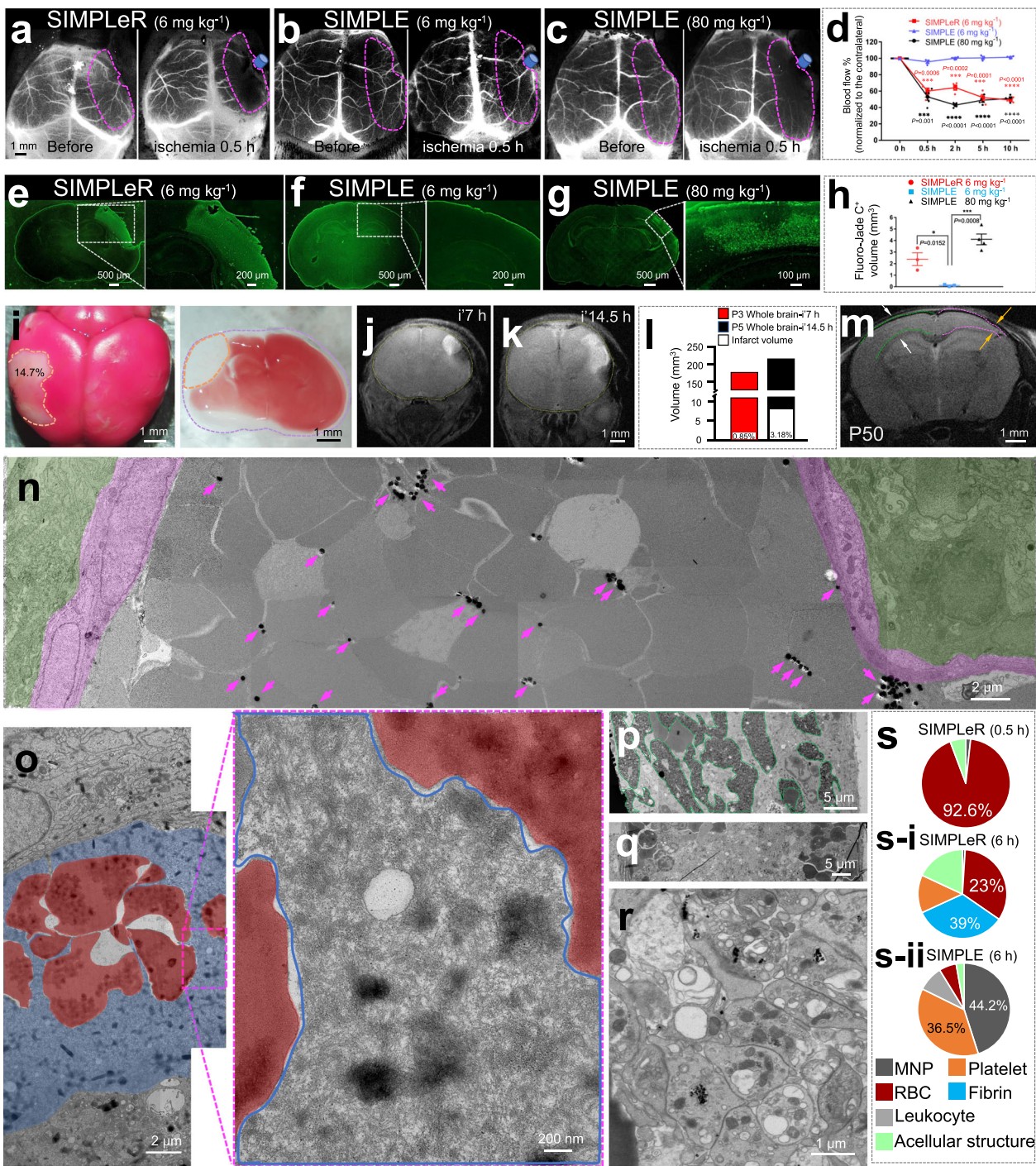

emerged and accounted for one-sixth at 6–10 h following occlusion (Fig. 4f). Thus, there was a positive relationship between the hemorrhagic transformation rate and ischemia duration within hours (Fig. 4f). In contrast, hemorrhagic transformation in adult mice usually occurs during the subacute stage within days[30]. These results suggested that hemorrhagic transformation in human PAIS patients might be underestimated because it might occur at the acute ischemic stage, which is more difficult to detect and diagnose.

At 6 h of occlusion, more severe hemorrhagic transformation took place in rat pups than in mouse pups, as evidenced by the 89% prevalence of parenchymal hematoma PH-type transformation in rats versus 17% in mice (Supplementary Fig. 7a, c 'vehicle group' versus Fig. 4e-6h, f-6h). These results

demonstrated that the phenomenon of hemorrhagic transformation after brain ischemic stroke is evolutionary conserved across mice, rats, and humans. Thus, PAIS modeled in rodent animals by SIMPLeR and SIMPLE mimicked the clinical observations of ischemia-induced hemorrhagic transformation, and they are reliable models that merit broad applications in the basic research field of PAIS.

There have been very few comparative epidemiology studies that systematically analyze the age-dependent incidence of hemorrhage transformation after focal cerebral ischemia across the age categories of perinatal fetuses, preterm and full-term infants, children, and adults. We studied the hemorrhagic transformation rate, specifically at 6–10 h occlusion, crossing the different ages restricted to P4, P15, and P80 mice. We

**Fig. 3 Comparison between SIMPLeR and SIMPLE. a–d** Less MNPs in the SIMPLeR territory than in the SIMPLE-induced hypoperfusion in the dMCA territory. **a, b** Representative LSCI of two live P3 mouse pups at 0.5 h of SIMPLeR and SIMPLE using the same MNP dose (6 mg kg$^{-1}$). (**c**) LSCI images of a P5 mouse pup insulted by 0.5 h of SIMPLE using 80 mg kg$^{-1}$ MNPs. **d** Quantification of the time course of cerebral blood flow change in the ischemic hemisphere normalized to that of the contralateral side for **a, b**, and **c** ($N = 4$ mice for each time point). For each time point, the effect sizes of SIMPLeR versus SIMPLE (6 mg kg$^{-1}$) are 12.9, 26.4, 18.5, and 38.4 and those of SIMPLE 80 mg kg$^{-1}$ versus 6 mg kg$^{-1}$ are 15.8, 42, 19.7, and 39.2. **e–g** Representative images of Fluoro-Jade C staining of brain sections from P3 mouse pups insulted with SIMPLeR and SIMPLE. **h** Quantification of the volume of Fluoro-Jade C -positive brain regions for **e, f**, and **g** ($N = 4$ mice per group). The effect sizes of SIMPLeR and SIMPLE (80 mg kg$^{-1}$) relative to the reference group SIMPLE (6 mg kg$^{-1}$) are 34.3 and 60.3, respectively. (**i**) Macroscopic images of TTC staining of the whole brain (P6) and brain sections (P4) from mouse pups after 10 h of focal ischemia. T2W MRI of P3 mouse pups at 7 h (**j**) and P5 at 14.5 h (**k**) after SIMPLE. **l** Quantification of infarcted brain volume for **j** and **k**. **m** The second MRI T2W image for the same mouse in **j** at P50. Pink and green dashed lines indicate ischemic ipsilateral and contralateral cerebral cortex, respectively. **n** Stitched TEM images of the dMCA segment shown in Fig. 2f, g, and Supplementary Fig. 5. Magenta arrows show MNPs bound to RBCs; vascular wall layers are marked in magenta and green. **o** Stitched TEM image of a red embolus located at the dMCA of P3 mouse pups that underwent 6-hour SIMPLeR. RBCs are marked in red, and surrounding fibrin is marked in blue. Blue line outlines the edge of fibrin in Insert. **p–r** Stitched TEM images of the dMCA segment (Supplementary Fig. 6) of P3 mouse pups with SIMPLE. Aggregated MNPs (180 nm) are outlined by green lines in **p**. The enriched white blood cells and platelets adjacent to the MNP occlusion are shown in **q** and **r**, respectively. **s-s-ii** Statistical analysis of different components of white thrombus-like obstructions or red emboli for **n–r**. A total of 766 μm$^2$ (**n**), 319 μm$^2$ (**o**) and 537 μm$^2$ (**p, q, r**) TEM images were analyzed. *$p < 0.05$, ****$p < 0.0001$, *ns* not significant. Data are presented as Mean ± SEM.

discovered that there was a neat, negative correlation between brain maturation status and hemorrhagic transformation rate (Fig. 4g, h). The more mature the brain is, the lower the hemorrhagic transformation rate at the acute phase. This finding seems to echo the findings in humans that the hemorrhagic transformation rate is 5-fold higher in children than in adults[21,33].

Our previous publication implied that damaged SMCs are cleared by migrating macrophage-like cells[26]. Additionally, it is well known that the removal of mural cells, including SMCs, leads to vascular disruption and bleeding[34,35]. We hypothesize that macrophages are involved in hemorrhagic transformation occurrence after ischemic insult. Minocycline, a macrophage inhibitor, ameliorated the severity of microbleeds (Supplementary Fig. 7a–c). Our experiment showed that 58% of rats in the minocycline-treated group had parenchymal hematoma (PH) type (7/12 rats), compared to 89% of rats in the vehicle-treated group (8/9 rats, Supplementary Fig. 7c). Alternatively, we used the strategy of macrophage deletion to further test our hypothesis. As previously reported, long-term inhibition of colony-stimulating factor 1 receptor (CSF1R) by feeding PLX 5622 for 2 weeks resulted in nearly full depletion of circulating monocytes/macrophages and brain resident macrophages, including microglia and border associated macrophages, in mice[36]. Notably, hemorrhagic transformation was completely prevented by this pretreatment in mouse pups (Fig. 4i, j). We further found that the rate of hemorrhagic transformation was tightly and reversely correlated with the residual macrophage density in the cerebral cortex (Fig. 4k). Two weeks of feeding PLX 5562 was less potent in rats than in mice (Supplementary Fig. 7d–f). In rats, PLX 5622-treatment did not decrease the overall rate of hemorrhagic transformation but significantly ameliorated the severity of microbleeds (Supplementary Fig. 7g–i). Taken together, our findings demonstrate that CSF1R-positive macrophages, including but not limited to microglia, are the crucial cell types that mediate focal cerebral infarction-induced hemorrhagic transformation.

## Discussion
As a method to establish an embolic stroke model in perinatal mice, SIMPLeR has several strengths over the traditional approaches. First, SIMPLeR can induce in situ embolization by aggregating red blood cells in vivo. This cannot be achieved by conventional strategies in which a red embolus is pre-formed in vitro[37]. The vascular microenvironment for embolus formation

in SIMPLeR is more natural than the plastic cannular tube used for embolus induction in the classical approaches. Although the photothrombotic stroke model can also induce de novo occlusive obstruction in vessels, the photoactive dye (Rose Bengal) results in a pure platelet occlusion that has few RBCs[16], which, however, is out of scope of modeling a red embolic stroke. Second, SIMPLeR requires no surgical operations on arteries or skulls. These operations are indispensable in traditional strategies that require transarterial lodging of the embolus or local injection of thrombin after opening a cranial window on the skull, which may cause a secondary injury. More importantly, it is extremely challenging to implement these surgical operations on perinatal mice that have light body weights (1–3 g). The third strength of SIMPLeR is to reversibly occlude vessels by precisely controlling the occlusion duration, whereas conventional approaches either produce permanent occlusion or unpredictable spontaneous reperfusion[38]. Overall, embolus introduction and final lodgment of the embolus within the cerebral vasculature make the previously established embolic models difficult to manage, leading to variations in infarct size and affected brain regions[39]. SIMPLeR can address all these difficulties by controlling embolism with spatiotemporal precision.

In addition, SIMPLeR demonstrated its superiority over SIMPLE in mimicking embolic stroke, while it preserved the strengths of SIMPLE, such as reversibility and the ability to generate the model without operating surgically on the skull or arteries. First, SIMPLeR is more clinically relevant than SIMPLE because the former relies almost exclusively on biological materials such as RBCs and fibrins to cause stroke and the latter utilizes mechanical occlusion with MNPs. Second, SIMPLeR uses much fewer MNPs than SIMPLE because RBCs are thousands of times larger than the nanoparticles we used. Using a low dose of MNPs is of great importance because we found that injection of an extremely high dose of MNPs between 150–200 mg kg$^{-1}$ caused significant lethality in adult mice, demonstrating that injecting an appropriate MNP dosage is crucial. We also found MNP accumulation in perinatal mouse internal organs following SIMPLE but not in SIMPLeR (Supplementary Fig. 3; Movies 8–11). The drawback of both SIMPLE and SIMPLeR is that neither of them can induce infarction in the deep brain regions because of the limitation of penetration depth of the magnetic field.

Hemorrhagic transformation is a natural evolution of cerebral infarction and causes secondary brain damage[20,21]. Both humans and adult rodents manifest hemorrhagic transformation at the subacute stage after ischemic stroke[30,33], but when hemorrhagic transformation takes place in PAIS remains unclear. Our study

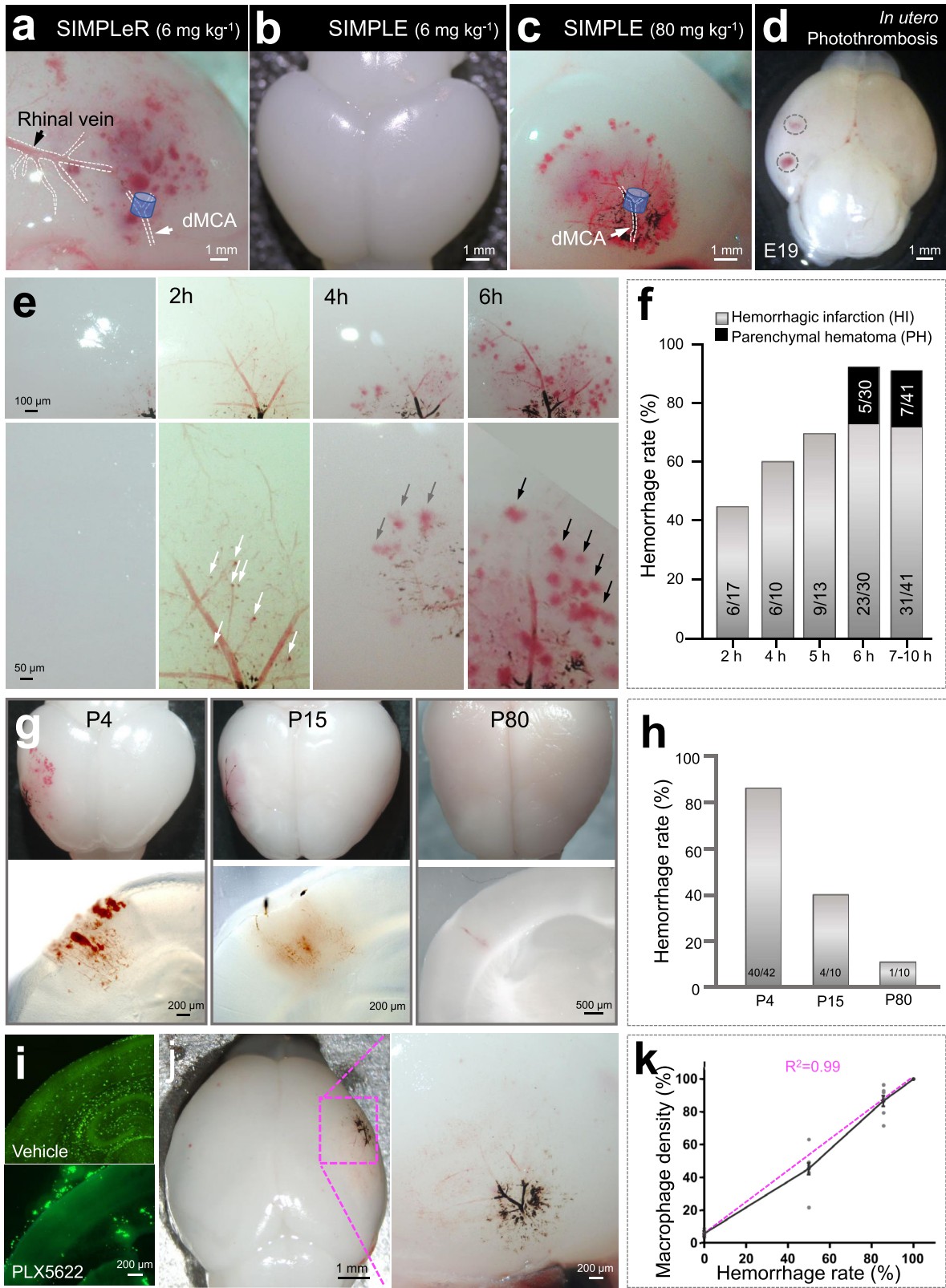

implies that PAIS patients may have hemorrhagic transformation at a very early time window following ischemia onset. There is controversy regarding the possibility of a macrophage-dependent mechanism mediating hemorrhagic transformation in adult ischemic stroke[24,30]. This study clearly demonstrated that inhibition or depletion of macrophages remarkably prevented hemorrhagic transformation in mouse and rat pups at postnatal days 3 to 5 after arterial occlusion by showing a linear relationship between macrophage density and hemorrhagic transformation probability (Fig. 4). However, it is still not clear whether circulating macrophages or brain resident central nervous system (CNS) macrophages play a major role, which merits further investigation. Hence, our research demonstrated that PAIS may have distinct early pathology from adult ischemic stroke, which,

**Fig. 4 Both SIMPLeR and SIMPLE resulted in hemorrhagic transformation in neonatal mice. a–c** Macroscopic pictures of brains from P3-5 mouse pups that underwent 6-hour SIMPLeR or SIMPLE at 6 or 80 mg/kg MNPs. **d** Picture of a brain from embryonic day 19 mice that underwent *in utero* photothrombosis at E18 taken by dissecting microscopy. **e** Representative bright-field images of brains of P2-7 mouse pups that underwent different occlusion durations ranging from 2 h to 6 h. **f** Quantification of the hemorrhagic transformation rate after different occlusion durations. Gray indicates the mild HI type, while black indicates the more severe PH type of hemorrhagic transformation. $N = 17$ to 41 mouse pups for each group. **g** Representative pictures of mouse brain and brain sections from P4, P15, and P80 mice that underwent 6–10 h of SIMPLE (P4 and P15) or MCA occlusion(P80). **h** Quantification of the hemorrhagic transformation rate for **g**. **i** Confocal images of the CNS macrophage marker Iba1 in brain sections from vehicle- or PLX 5622-treated mouse pups. **j** Macroscopic picture of a brain from a P3 mouse pup that had long-term PLX 5622-treatment prior to 6 h of ischemic stroke. **k** The reverse correlation of CNS macrophage density in the brain cortex and hemorrhagic transformation rate at 6 h of ischemic occlusion. The magenta dashed line represents the trendline. Data are presented as Mean ± SEM.

in turn, suggests the need for a specialized treatment strategy for PAIS.

## Methods

**Animals**. All animal experiments were carried out in accordance with protocols approved by the Institutional Animal Care and Use Committee (IACUC) at the School of Life Sciences, Westlake University. Wild-type *C57BL6/J* mice were purchased from the Laboratory Animal Resources Center of Westlake University. Sprague Dawley rats (250–290 g) were purchased from SLAC Laboratory Animal Co., Ltd. (Shanghai, China). Standard chow and water were provided to mice *ad libitum*. Four mice or two rats were housed in each cage. All animals were housed in a standard animal room with a 12/12-hour light/dark cycle at 25 °C. Both male and female mice and rats were used in this study.

**Synthesis of SiO2-coated Fe₃O₄-core MNPs**. To obtain MNPs, 0.75 g of FeCl₃•6H₂O was fully dissolved in 25.0 mL of ethylene glycol, followed by the addition of 0.18 g of sodium citrate tribasic dihydrate. After the complete dissolution of sodium citrate tribasic dihydrate, 1.2 g of sodium acetate was added to the above solution. After vigorous stirring for 0.5 h, the as-prepared mixture was slowly poured into a Teflon-lined stainless-steel autoclave reactor (capacity: 50.0 mL) and heated at 200 °C for 10 h. After the surface temperature of the autoclave cooled to room temperature, the reactor was carefully removed. Then, the as-prepared products, which were black in color, were thoroughly washed with ethanol and deionized water several times to eliminate the byproducts and finally dispersed in deionized water for further usage.

**Synthesis of streptavidin-MNPs**. To prepare Fe₃O₄@SiO₂ nanoparticles, we began by fully dispersing 50.0 mg of MNPs in a solution consisting of 160.0 mL of ethanol, 40.0 mL of H₂O, and 4.0 mL of aqueous ammonia (25~28 wt%). The mixed solution was treated by ultrasonication at 25 °C for 30 min. Then, 2.0 mL of tetraethyl orthosilicate (TEOS) was gradually added by injection to the well-dispersed solution with mechanical stirring at 25 °C. After 6 h, the obtained Fe₃O₄@SiO₂ nanoparticles were washed with ethanol and deionized water several times to remove byproducts. To prepare sulfhydryl-functionalized MNPs, the necessary quantity of Fe₃O₄@SiO₂ nanoparticles were immersed in 3-mercaptopropyltrimethoxysilane (MPTMS) solution (4% in anhydrous ethanol) and stirred for approximately 1 h at 30 °C. Next, the modified nanoparticles were washed with ethanol and dimethyl sulfoxide (DMSO) to remove the uncombined MPTMS. Then, the N-succinimidyl 4-maleimidobutyrate (GMBS) solution (7.0 mM in DMSO) was added to the dispersed nanoparticle solution and allowed to react for approximately 45 min at 30 °C. After the reaction, the MNPs were washed with DMSO and phosphate-buffered saline (PBS) to remove the excess GMBS. Subsequently, the modified MNPs were dipped in streptavidin ($50.0 \mu g \, mL^{-1}$ in PBS) for 2 h at room temperature to coat their surfaces with streptavidin. Finally, the mouse RBC-specific biotinylated anti-Ter119 monoclonal antibody was incubated with streptavidin-MNPs at room temperature in a shaker for 2 h to obtain antibody-functionalized MNPs. Some of the SIMPLeR experiments used streptavidin-modified, PEG-2000-coated MNPs (Cat# Mag9301-05 Beijing Zhongkeleiming Daojin Technology Co., Ltd.)

**Procedure for generating biotin-mRBCs**. Allogenic peripheral blood from young and healthy adult WT mice (4-6 weeks) was collected and washed with sterile PBS. Forty million RBCs were incubated with biotin-conjugated anti-Ter119 antibody (Cat# 13-5921-82, Thermo Fisher Scientific) diluted at the concentration of $10 \mu g \, mL^{-1}$ on ice for 30 min and then washed and centrifuged at 200 g for 10 min at 4 °C. Next, these biotin-RBCs^Ter119 were mixed with streptavidin-MNPs at various doses as indicated in the **Results**. Streptavidin-MNPs with an average size of 300 nm were produced by our collaborators at Westlake University (Dr. Botao Ji's laboratory and Dr. Bobo Dang's laboratory). The mixture of biotin-RBC^Ter119 and the streptavidin-MNPs was kept on a shaker for another 20 min, and then the procedure of generating magnetized biotin-mRBCs was completed. Unmagnetized RBCs underwent the same treatment, but without streptavidin-MNPs. For each single step, the RBC quality was closely monitored by microscopic inspection. Once

the erythrocytes that lost a biconcave shape accounted for more than 10%, we ceased the process and initiated a separate mRBC preparation.

**Procedure for generating mRBCs using Lipofectamine**. Allogenic peripheral blood was collected from healthy WT mice in Hanks' balanced salt solution (HBSS, Cat# 14175-095, Gibco). RBCs were washed twice with HBSS, and a final cell concentration of approximately $4 \times 10^8 \, ml^{-1}$ was achieved. Each magnetization reaction consisted of two mixtures: Mixture A consisted of 5 μl Lipofectamine 3000 (Cat# L3000075, Invitrogen) and 50 μl HBSS; Mixture B was composed of 50 μl HBSS and polyethylene glycol (PEG)-2000-coated MNPs (Cat# 09-82-182 S13218, Nanomag-D, PEG-2000, 180 nm, 10 mg/ml, MicroMod, Germany) at different dosages. Each mixture was kept alone for 5 min at room temperature (RT) and then mixed, and then this final mixture was kept at RT for another 30 min before being incubated with RBCs. This final magnetization system was kept at 37 °C for 3 h and then completed.

**mRBC enrichment assay (mREA)**. We created the following mREA means for statistical analysis of magnetization efficiency. A drop of 20 μl RBC-nanoparticle mixture, including unmagnetized RBCs and mRBCs, was placed on the foil with a rectangular magnet underneath (length x width x height = 5 cm × 2 cm x 0.5 cm). The center of the liquid drop was aligned with the edge of the magnet. Upon this magnetic gradient, mRBCs moved into a line that aligned with the edge of the magnet, whereas unmagnetized RBCs did not move (Fig. 1g-i, g-ii). A similar principle, as described above, of quantifying RBC magnetization efficiency by mREA is to use a 1-well magnet frame (Cat# 18000, EasySep™ Magnet), which is a piece of chemistry equipment consisting of a magnetic bead separation tool, to separate mRBCs and unmagnetized RBCs. We calculated the percentage of unmagnetized RBCs compared to the total number of RBCs we used. Finally, the mRBC percentage was determined by subtracting the unmagnetized RBC percentage. Thus, the magnetization efficiency was quantified.

During the RBC magnetization reaction, RBCs were sometimes prelabeled either with lipophilic fluorescent dye DiO (Cat# V22886, Invitrogen) or with Alexa Fluor 488-conjugated streptavidin. Therefore, in principle, both unmagnetized RBCs and mRBCs were visualized under a fluorescence microscope.

**Scanning electron microscopy (SEM) for RBCs**. RBCs were fixed with 2.5% glutaraldehyde solution and then postfixed with 1% osmic acid for another 1.5 h. After each fixation, the samples were washed three times with 0.1 M, pH 7.4 PBS for 15 min. The samples were dehydrated in increasing concentrations of ethanol (30, 50, 70, 80, 90, 95, and 100%) for 15 min at each step. The samples were dried and coated with aurum before being viewed either with a Nova 450 or with a Zeiss Gemini 550 scanning electron microscope or a Nova 450 scanning electron microscope.

**Transmission electron microscopy (TEM)**. The unmagnetized RBCs, mRBCs, and occluded distal MCAs were fixed as described above and contrasted with 2% uranyl acetate for 30 min at room temperature. The samples were gradually dehydrated by successive baths in 50%, 70%, 90%, and 100% ethanol. Next, the samples were washed twice with 100% acetone and embedded in resin Epon. Ultrathin sections at 60 nm thickness were obtained and deposited on copper grids to be examined with a Talos L120C G2 transmission electron microscope or Tecnai T10 transmission electron microscope.

**SIMPLeR-mediated dMCA occlusion ischemic stroke**. The cylindrical magnet (NdFeB) was 1 mm long and 1 mm in diameter. The properties of this magnet have been well characterized; its estimated magnetic force is in the range of 0.25–0.5 pN[26]. The magnet was glued onto the skull of the mouse pup, covering the dMCA, prior to the administration of the prepared mRBCs, at various amounts, to the bloodstream via the superficial temporal vein. Upon magnetic field application, mRBCs accumulated at the targeted vessels. Thus, SIMPLeR-mediated dMCA occlusion was achieved. The duration of ischemia and reperfusion onset timing was controlled by when to remove micromagnets.

**Laser speckle contrast imaging (LSCI)**. The theories and techniques of LSCI have been documented in the literature[40]. LSCI provides a measure of blood flow velocity by quantifying the extent of blurring of dynamic speckles caused by the motion of red blood cells through the vessels. Briefly, mice were anesthetized with isoflurane and placed under a RFLSI III device (RWD Life Sciences). The skull over both hemispheres was exposed by making an incision along the midline of the scalp. When a 785 nm laser was used to illuminate the brain, it produced a random interference effect that represented blood flow in the form of a speckle pattern. Scattering light was detected by a charge-coupled device (CCD) camera, and the images were acquired by custom software from RWD Life Sciences Company. For each acquisition, a total of 160 images, each of which measured $2048 \times 2048$ pixels, were collected at 16 Hz.

**TTC staining**. At 10 h after focal arterial ischemic stroke, the brains of mouse pups were dissected. Whole brain or 1 mm thick transverse brain sections were incubated with 2% TTC (Cat# A610558-0005, BBI) dissolved in PBS at 37 °C for 15 min. Next, brain sections were fixed with 4% PFA overnight. Images were taken with a fluorescence stereo zoom microscope (Zeiss Axio Zoom V16).

**Immunostaining**. Briefly, mice were euthanized by injection of an overdose of pentobarbital sodium anesthetic and fixed via cardiac perfusion with 4% paraformaldehyde (PFA). Brains were dissected out and postfixed in 4% PFA for another 2–4 h. Brain dehydration was achieved by soaking in sucrose gradients of 10%, 20%, and 30%. Brain cryosections at various thicknesses (20–60 μm) were obtained by using a cryostat (CM3050S, Leica). Immunofluorescent staining was performed using the following steps: 1. Permeabilization in 0.5% Triton X-100 diluted with PBS; 2. Blocking in buffer containing 5% BSA plus 0.1% Triton X-100 in PBS; 3. Primary antibody incubation at 4 °C overnight with antibodies against Iba1 (1:200, Cat# 019-19741, Wako), GFAP (1:400, Cat# PA1-10004, Thermo Fisher Scientific), and NeuN (1:400, Cat# ABN90P, Millipore); 4. Incubation at room temperature for 2 h with secondary antibodies conjugated with various fluorophores, such as Alexa Fluor 488, Alexa Fluor 555, and Alexa Fluor 647 (1:400, Cat# A–11001, A–11003, A–21235, Invitrogen); 5. Hoechst 33342 counterstaining by coincubation with the secondary antibody. The images were acquired with a Zeiss LSM800 confocal microscope.

**Fluoro-Jade C staining**. As previously described[41], Fluoro-Jade C detects neurons that are undergoing degeneration. Briefly, 20 μm brain sections were subjected to the following consecutive steps. (1) Rehydration and permeabilization in basic ethanol (1% NaOH in 80% ethanol for 5 min); (2) Transfer to freshly prepared 70% ethanol for 2 min and washing with distilled water for 1 min; (3) Blocking of background fluorescence and optimization of contrast by incubation in 0.06% potassium permanganate for 15 min; (4) Rinsing in distilled water; 5. Incubation of brain sections in Fluoro-Jade C working solution (0.001% Fluoro-Jade C in 0.1% acetic acid solution) for 15 min, followed by rinsing in distilled water three times for 1 min each. The slides were sealed with DPX (Sigma) mounting medium. All processes were completed at room temperature.

**T2-weighted MRI scans**. MRI experiments were conducted by a 7.0-tesla scanner (BioSpec 70/20 USR, Germany) using a transmit quadrature birdcage and a mouse surface array receive coil for in vivo measurements. T2-weighted (T2W) images of perinatal mice with stroke at different time points. The images were obtained using the following acquisition parameters: repetition time (TR)/echo time (TE) 3000 ms/36.47 ms, flip angle 50°,$15 \times 15$ mm$^2$ field of view, $256 \times 256$ matrix, 2 excitation, 20 slices, 0.5-mm contiguous slice thickness. The high signal region of MRI indicated the damaged brain region of mouse pups with stroke.

**Cranial imaging of mouse pups by two-photon microscopy**. Mouse pups were anesthetized with isoflurane, and analgesia was provided by intraperitoneal injection of 0.2% meloxicam. The scalp was removed, and the skull was exposed and cleaned. A dental drill with a 0.6-mm-diameter bit was used to engrave and thin the bone around the circular craniotomy area at a size of 3 mm. The piece of skull was carefully peeled off with fine-pointed forceps, and a cranial window was generated. A coverslip with a 3-mm diameter was placed on the cranial window, and its perimeter was completely sealed with a 1% agarose gel. The headplate was glued with dental acrylic, through which the mouse pup was fixed on a headplate holder. The headplate holder with a mouse pup was placed under a two-photon microscope (FVMPE-RS, Olympus). The bloodstream was visualized with FITC-dextran. All surgeries were performed with sterilized instruments and an environment.

**In utero photothrombosis**. The principle of photothrombosis induction was according to a previous report[42]. However, we modified the procedure to induce brain damage in mouse embryos. Rose bengal (80 mg kg$^{-1}$, 330000, Sigma) was intraperitoneally injected into a female mouse at gestational day 18. A pregnant mouse was anesthetized with isoflurane. The heads of the embryos were illuminated for 15 min by a 535 nm laser with 100 mW power. Sterile saline and wet gauzes were applied to avoid dryness of the uterus. Analgesia was provided by

intraperitoneal injection of 0.2% meloxicam. At 24 h after photothrombosis, the brains of the illuminated embryos were dissected out, and bright-field images were taken by a Stereo Zoom Microscope (Zeiss Axio Zoom. V16) to examine the hemorrhagic transformation. Sham control was conducted with the same procedure, except intraperitoneal injection of vehicle instead of rose bengal.

**MCA occlusion**. Focal cerebral ischemia was induced by MCA occlusion as described previously[26]. Briefly, adult mice (P80) were anesthetized with sodium pentobarbital (80 mg kg$^{-1}$) and placed on a heating pad to maintain body temperature. Under a dissecting microscope, the right common carotid artery and external carotid artery were exposed via an incision along the midline of the neck. A filament (Cat# 602356PK5, Doccol) was inserted into the lumen of the carotid artery and advanced to the MCA. The occlusion lasted for 6 h. Mouse brains were dissected for hemorrhagic transformation evaluation.

**Macrophage depletion with PLX5622**. Macrophage depletion was achieved by administering the CSF1R inhibitor PLX 5622, as previously described[36,43]. The pregnant dams starting at embryonic day 6 (E6) were fed a PLX 5622-formulated AIN-76A diet (1.2 g PLX 5622 per kilogram of diet, Plexxikon) ad libitum. Control dams received a control diet (AIN-76A, Research Diets) accordingly. After mouse pups were born, the diet was continuously provided to mother mice for an additional 2–3 days when perinatal arterial ischemic stroke was modeled.

**Macrophage activity inhibition with minocycline**. Minocycline (45 mg/kg, Cat# M9511, Sigma)[24,44] was intraperitoneally injected into neonatal SD rats (P2 - P4) twice, at 1 h prior to and at 3 h after SIMPLE onset. At 6 h after SIMPLE, the rat pup brains were dissected for hemorrhagic transformation evaluation.

**Statistics and Reproducibility**. The quantified data in all figures were analyzed with GraphPad Prism 7.0 (La Jolla, CA, USA) and presented as the mean ± SEM with individual data points shown. Unpaired two-tailed Student's $t$-test was used for assessing the statistical significance between two groups. Statistical significance was determined by calculation of $p$-value (*$p < 0.05$, **$p < 0.01$, ***$p < 0.001$, and ****$p < 0.0001$, ns: not significant). The absolute values of effect sizes are reported in this study. The repetition of our data is independent biological replicates, the number of replicates for each experiment noted in the corresponding figure legend.

**Reporting summary**. Further information on research design is available in the Nature Research Reporting Summary linked to this article.

## Data availability
The source data underlying the graphs and charts shown in the figures and tables are provided in Supplementary Data 1. All data generated or analyzed during this study are included in this published article (and its supplementary information file).

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

## Acknowledgements

We thank W.Z.S. for advising us on live imaging of mouse pups; Y.W., L.G., G.F., and J.G. at the Westlake University Microscopy Core Facility for their technical support with TEM and SEM imaging; D.S. and P.Y. at the Center of Cryo-Electron Microscopy (CCEM), Zhejiang University, for their technical assistance with TEM and SEM; F.X. and Y.G. for their advice and assistance with light microscopy imaging; X.L. for critical discussion of the logical flow of the project; Z.Z. for providing insights based on a wealth of clinical knowledge; D.Z., J.L., T.L., X.H., J.R., and L.Z. at Westlake University for contributing to in-depth discussions of this project; and the animal facility for its technical assistance with rodent housing. This work is supported by the National Natural Science Foundation of China (31800864 and 31970969 to J.-M.J, 32170964 to W.-p.G., 22077104 to B.D.), Westlake startup funds, Westlake Education Foundation and MRIC funds (103536022011) to J.-M.J.; Westlake Education Foundation to B.D.; and a Zhejiang Province Natural Science Foundation of China grant (LQ19H090009) to X.X.

## Author contributions

J.-M.J. and W.-p.G. conceived the project, and Y.J. performed most of the animal experiments and analyzed the data. B.C. and J.Z. conducted some experiments on RBC magnetization in vitro. P.S., J.L., M.G., Y.H., X.G., B.D., and B.J. helped design and create one type of mRBCs. Y.G. performed ischemic stroke in adult mice. Y.W. conducted in utero photothrombosis. X.Z. performed and J.X. supported T2W MRI imaging. F.W. helped quantify the tissue infarction volume of MRI data. X.X. provided helpful guidance to Y.J.

## Competing interests

The authors declare no competing interests.
