## [Transparent Peer Review File · Communications Biology]

Reviewers' comments:

Reviewer #1 (Remarks to the Author):

In this manuscript, titled "Precise control of embolic stroke with magnetized red blood cells", Dr. Jia and co-authors reported a new embolism animal model based on SIMPLE technology, which the same group developed. The magnetized red blood cells were able to be reversibly manipulated to occlude cerebral vessels. The authors showed that both SIMPLE and SIMPLER could induce age-dependent micro-hemorrhage transformation at the acute ischemic stage. Additionally, microglia/macrophage depletion prevented the hemorrhage transformation in the SIMPLER ischemic model. The study is well organized and easy to follow. However, I have several concerns as below:

1. What is the novelty of this study? Compared to SIMPLE, what is the advantage of SIMPLER? What is the rationale to investigate micro-hemorrhage transformation in this model? I hope the authors could emphasize these aspects in the abstract.
2. As this model is mainly focused on mimicking the ischemic stroke in perinatal newborns, the authors should provide more background on perinatal ischemic stroke, but not an adult one. The red blood clot produced here cannot simulate the clot developed in the adult brain.
3. It is also necessary to explain the rationale to investigate micro-hemorrhage transformation and microglia/macrophage depletion in the introduction.
4. As an animal model, what are the criteria to determine the animals are qualified for further studies? And what about the success rate and survival rate?
5. The authors describe this model as a reversible cerebral blood vessel occlusion model. However, none of the data showed this "reversible process". And the occlusion vessels including but are not limited to microvessels.
6. On Page 7, "Thus, SIMPLER is more advanced than SIMPLE by using significantly less amount of MNP, indicating SIMPLER is more suitable than SIMPLE regarding long-term study of brain recovery after ischemia." How to interpret the correlation between less amount of MNP and the long-term study?
7. Is there any requirement for the RBC quality? And how to guarantee the purity of RBC for mRBC preparation?
8. Cerebral ischemia and the following hemorrhagic transformation can induce the infiltration of macrophages into the brain parenchyma. However, PLX5622 will deplete both microglia and macrophages. Minocycline is also non-specific. Thus, the authors need to revise the last part of the result section.

Reviewer #2 (Remarks to the Author):

The manuscript is well structured. The author presents and fully discusses the experimental methods for control of embolism, and the data and results are also comprehensive and clear. It is particularly commendable that the author preliminarily explores possible ways to prevent hemorrhage. Therefore, I recommend that this manuscript can be accepted after major revision as follows:

There are sloppy grammatical and editorial issues in this paper. For example, grammatical mistake in "The precise manipulation of embolism in rodent microcirculation can facilitate mechanistic studies of brain damage and repair after embolic stroke," "Embolism and thrombosis are responsible for over 80% of strokes, the most common cause of adult disability and the 2nd leading cause of death worldwide" is ambiguous. Similar errors appear many times. Many of them could cause difficulties to the readers. English should be improved in this manuscript.

Reviewer #3 (Remarks to the Author):

Dear authors,

Your manuscript "Precise control of embolic stroke with magnetized red blood cells" is a smart approach on reversibly induce an embolic stroke. RBCs are loaded with magnetic nanoparticles, such that they react on the magnetic field of a permanent magnet. At the location of the magnet an occlusion can be induced. You show a large variety of different analytical experiments to demonstrate and validate your findings. Please find below some comments and suggestions.

General:

- Avoid using abbreviation SIMPLE and SIMPLER in the abstract
- Please check if you have introduced all the used abbreviations (e.g. MCA?, SMS, CNS, ...) . In my opinion you are using abbreviations very extensively, which makes it sometimes hard to read. Think about not abbreviating everything which you are just using once or twice.
- Just as an idea, which came to my mind: why not using tomographic imaging, such as MRI, which could detect the MNPs as contrast agent? Maybe you could comment or discuss on this?

Introduction

- Fig 2 is introduced before Fig1
- When you introduce the abbreviation SIMPLER it does not fit the grammar of the sentence.
- You use the word "SIMPLE" before introducing it. The reader just knows that this is the technique done by you before.

Results

- P.4 | 94: "a drop of liquid magnet field gradient". That is not understandable, you mean something else (which becomes clear when reading the methods). Please rephrase.
- P4 | 100: how did you measure the 90s? I suppose the MNPs move and align gradually with the edge of the magnet. When did you stop measuring the time, as soon as you see a line? Or when no MNPs are moving anymore?
- P.6 | 140 (not only here) Why is the magnet you are using a micro-magnet? It is 1mm in diameter (this information should also be mentioned in the methods, I only found it in the figure caption). See my comment about missing information on the used magnets.

Discussion

- Dosis limit of MNP: you state "avoid potential longterm toxicity". Can you give a concentration limit when it becomes toxic and what happens then?
- Could the agglomeration of uncoated MNPs be a problem, when they are not longer attached to the RBCs?
- One drawback of your method is, that only vessels close to surface can be used, because of imaging and because of penetration depth of the magnetic field. Please comment on that and/or discuss possible solution strategies.
- Even though small magnet was used, the magnetic field may also affect vessels nearby. Did you observe that? Could it be a problem?

Methods

- Size and strengths of used magnets are missing
- I could not find all information on every used MNPs. Please try to give comparable information on the used particles.

Overview:

We thank all the reviewers for their appreciative comments, recognizing the importance and solidity of our work. We are grateful for the specific comments raised. We have performed additional experiments and data analysis. Each of question have now been addressed through providing the revised text, new supplementary figures, and new movies. Detailed responses are delivered through a point-by-point response letter.

We highlight the key revisions in this letter as below.

The major changes to the manuscript:

1. We have revised text throughout the manuscript, including grammatical, editorial, information missing, and abbreviation overuse problems.
2. We have included additional rationales and discussions in the INTRODUCTION and DISCUSSION sections respectively.
3. We have set up the specific criteria for SIMPLeR as an animal model.
4. We provide a new movie that shows that the reversibility of mRBC aggregates.

Reviewers' comments:

Reviewer #1 (Remarks to the Author):

In this manuscript, titled “Precise control of embolic stroke with magnetized red blood cells”, Dr. Jia and co-authors reported a new embolism animal model based on SIMPLE technology, which the same group developed. The magnetized red blood cells were able to be reversibly manipulated to occlude cerebral vessels. The authors showed that both SIMPLE and SIMPLER could induce age-dependent micro-hemorrhage transformation at the acute ischemic stage. Additionally, microglia/macrophage depletion prevented the hemorrhage transformation in the SIMPLER ischemic model. The study is well organized and easy to follow. However, I have several concerns as below:

1. What is the novelty of this study? Compared to SIMPLE, what is the advantage of SIMPLER?

We thank the reviewer for raising this point. We have provided multiple lines of evidence demonstrating that SIMPLER is more advanced than SIMPLE in terms of establishing an embolic stroke model. **First**, SIMPLE cannot use red blood cells (RBCs) as the main occlusive substance to obstruct the vessels, whereas SIMPLER can. Red blood cells and fibrin are known to be the major components of emboli. Following SIMPLER, TEM data clearly showed that RBCs constituted 92.6% of the embolus that was formed within 0.5 hours (Fig. 3N and S); furthermore, RBCs and fibrin together made up 62% of the occlusive material at 6 hours following SIMPLER; in contrast, SIMPLE relied on platelets (36.5%) and magnetic nanoparticles (44.2%) as the major obstructive substances, meaning that the resulting clots were not representative red emboli. **Second**, SIMPLER is superior to SIMPLE in that the former requires only one-thirteenth as many magnetic nanoparticles (MNPs) as the latter (Fig. 3A, C and D). As we clearly presented in Figure 3A and 3B, 6 mg/kg MNPs failed to block local blood flow in the SIMPLE model, whereas it succeeded in the SIMPLER model (Fig. 3A, B and D). Correspondingly, SIMPLER caused neuronal degeneration, whereas SIMPLE did not (Fig. 3E, F and H). Using a low dose of MNPs

is of great importance because we found that venous administration of an extremely high dose of MNPs (in this study, 150-200 mg/kg) caused significant lethality in adult mice, demonstrating that injecting an appropriate MNP dosage is crucial. In SIMPLE, although mouse pups that received a medium dose of MNPs (80 mg/kg) survived and remained in very good condition throughout their lives, we found MNP accumulations in pups' internal organs, such as the liver and spleen, at 6 hours after injection (Fig. S2 and the revised Movie 10 and Movie 11). However, in the SIMPLER model, MNP deposits in these organs were not detectable (Fig. S3 and the revised Movie 8 and Movie 9). Thus, SIMPLER avoids potential side effects, whereas SIMPLE may not. Overall, in terms of building an embolic stroke model, the SIMPLER model was superior to SIMPLE because the former was more relevant to the clinically observed emboli and used a smaller dose of MNPs than the latter.

In terms of the simplicity of stroke induction regardless of the main occlusive material, SIMPLER is more time consuming than SIMPLE because the latter does not require complicated mRBC preparation. Otherwise, SIMPLER is more advanced.

2. What is the rationale to investigate micro-hemorrhage transformation in this model? I hope the authors could emphasize these aspects in the abstract.

We thank the reviewer for this important suggestion. We revised our abstract and added our rationale (p. 2, lines 29-31). Additionally, please see the following revision: "Microhemorrhagic transformation is observed in one-third of infant patients who have suffered PAIS, but the underlying mechanism remains elusive." Related epidemiological studies can be found in the references¹⁻³ below.

3. As this model is mainly focused on mimicking the ischemic stroke in perinatal newborns, the authors should provide more background on perinatal ischemic stroke, but not an adult one. The red blood clot produced here cannot simulate the clot developed in the adult brain.

We agree with the reviewer and have modified our introduction by providing additional background on perinatal arterial ischemic stroke (p. 2, lines 41-53).

4. It is also necessary to explain the rationale to investigate micro-hemorrhage transformation and microglia/macrophage depletion in the introduction.

We thank the reviewer for this suggestion; we have added our rationale to the INTRODUCTION section (p. 3, lines 62-78)

5. As an animal model, what are the criteria to determine the animals are qualified for further studies? And what about the success rate and survival rate?

The reviewer has raised critical points. The **first** criterion was the presence of a visible red thrombus in the targeted vessels by visual inspection under a stereomicroscope when the magnet was removed. The **second** criterion was the extent of reduction in regional cerebral blood flow in the middle cerebral artery territory. When cerebral blood flow in the affected hemisphere, as quantified by laser speckle contrast imaging, was reduced to 40-50% of the contralateral value (Fig. 3A-D), neuronal death and brain tissue infarction were consistently detected by Fluoro-Jade C staining and TTC staining, respectively (Fig. 3E-I). Thus, the gold standard criterion was a sufficient reduction in regional blood flow measured by laser speckle contrast imaging, which directly indicated a successful stroke model; animals that fulfilled this criterion were qualified for further study. In addition, rodent pups' skin color and respiration rate were closely monitored during anesthesia and surgery.

The success rate of red thrombus formation in the distal middle cerebral artery was 100%, but the success rate of stroke induction was 69% because not all occlusions necessarily resulted in ischemia. The survival rate was 100%, as all the pups that were subjected to the SIMPLeR model survived. We have included this statistical analysis in our revised article (p. 8, lines 212-215) and Supplementary Figure 4E.

6. The authors describe this model as a reversible cerebral blood vessel occlusion model. However, none of the data showed this “reversible process”. And the occlusion vessels including but are not limited to microvessels.

We agree with the reviewer that the occlusion induced by SIMPLeR and SIMPLE included both

microvessels and larger arterioles and arteries when we specifically aimed to occlude the distal middle cerebral artery in this study. We have rephrased the statement in our revised manuscript.

We demonstrated the reversibility of the process through a test at the venous sinus, shown in Fig. 2E. We occluded the venous sinus for a very short period (9 seconds) and found that the majority of the magnetized blood cells (green) flew away at the 38th second. Please compare the abundance of mRBCs (green dots) between the last panel and the middle panel of Fig. 2E (indicated by pink arrows). However, we acknowledge that we did not verify the reversibility at the distal middle cerebral artery (MCA). **In this resubmission, we provide new data showing that the red thrombus at the distal MCA was dispersed once the magnetic field was removed** (p. 7, lines 179-181, see the revised Movie 6).

7. On Page 7, “Thus, SIMPLeR is more advanced than SIMPLE by using significantly less amount of MNP, indicating SIMPLeR is more suitable than SIMPLE regarding long-term study of brain recovery after ischemia.” How to interpret the correlation between less amount of MNP and the long-term study?

We apologize for causing confusion about this point in the previous manuscript. As mentioned before, we found MNP deposits in the liver and spleen following SIMPLE but not SIMPLeR. Moreover, we found that venous administration of an extremely high dose of MNPs, between 150 and 200 mg/kg, caused considerable lethality in adult mice, demonstrating that this extremely high dose of MNPs is toxic. Thus, we anticipated that the deposits formed in the SIMPLE model with a medium dose of MNPs (80 mg/kg) might cause long-term side effects on liver function, but we do not have any direct evidence thus far to show the long-term side effects. In our revised manuscript, we have rephrased our statement (p. 8, lines 194-196). At present, we have started to systematically evaluate the long-term side effects of MNPs on animal health in an additional project that is ongoing in our laboratory.

8. Is there any requirement for the RBC quality? And how to guarantee the purity of RBC for mRBC preparation?

We thank the reviewer for raising the issue. Yes, there is. To determine how much anti-Ter119 antibody (RBC marker antibody) we needed to add to the magnetization reaction system for mRBC preparation, we needed to quantitatively analyze the red blood cell density under the microscope. During this process, we also performed a microscopic assessment of red blood cell quality. We ceased the mRBC preparation process if we found that under 90% of erythrocytes had a normal biconcave shape. We have included this requirement in the revised Methods section (p. 15, lines 390-392). The quality of RBCs at each single step during the magnetization process is shown in the revised Supplementary Fig. 2 and below. Our results demonstrated that mRBC preparation via streptavidin-biotin binding did not cause serious damage to RBCs (Fig. 1B-E and images shown below)

Figure caption :

RBC quality monitoring at each single step during mRBC preparation. A. A representative bright-field image of the washed RBCs prior to the antibody incubation step. B. Scanning electron microscopy (SEM) image of RBCs coated with an antibody that specifically binds to RBCs (biotinylated anti-Ter119 antibody). C. SEM image of the final magnetized RBCs. D. Quantification of the percentage of RBCs with a biconcave shape in each step.

Quantification of the percentage of RBCs with a biconcave shape in each step.

As illustrated in Fig. 1A and 1A', magnetic nanoparticles modified with streptavidin can be specifically attached to RBCs via the biotinylated anti-Ter119 antibody, which recognizes RBCs but not white blood cells. Our SEM inspections of mRBCs *in vitro* (Fig. 1B-1E) and *in vivo* (Fig. 1F and Fig. 3N) merely detected any magnetized white blood cells. These results suggested that our mRBC preparation magnetized RBCs almost exclusively, with very few white blood cells included.

9. Cerebral ischemia and the following hemorrhagic transformation can induce the infiltration of

macrophages into the brain parenchyma. However, PLX5622 will deplete both microglia and macrophages. Minocycline is also non-specific. Thus, the authors need to revise the last part of the result section.

We thank the reviewer for this important comment. We have revised the last part of the RESULTS section (p. 11, lines 291-294, p. 12, lines 299-301).

References

1. Beslow, L.A. et al. Hemorrhagic transformation of childhood arterial ischemic stroke. *Stroke* 42, 941-946 (2011).
2. Cole, L. et al. Clinical Characteristics, Risk Factors, and Outcomes Associated With Neonatal Hemorrhagic Stroke: A Population-Based Case-Control Study. *JAMA Pediatr* 171, 230-238 (2017).
3. Hutchinson, M.L. & Beslow, L.A. Hemorrhagic Transformation of Arterial Ischemic and Venous Stroke in Children. *Pediatric Neurology* 95, 26-33 (2019).

Reviewer #2 (Remarks to the Author):

The manuscript is well structured. The author presents and fully discusses the experimental methods for control of embolism, and the data and results are also comprehensive and clear. It is particularly commendable that the author preliminarily explores possible ways to prevent hemorrhage. Therefore, I recommend that this manuscript can be accepted after major revision as follows:

There are sloppy grammatical and editorial issues in this paper. For example, grammatical mistake in "The precise manipulation of embolism in rodent microcirculation can facilitate mechanistic studies of brain damage and repair after embolic stroke," "Embolism and thrombosis are responsible for over 80% of strokes, the most common cause of adult disability and the 2nd

leading cause of death world-wide” is ambiguous. Similar errors appear many times. Many of them could cause difficulties to the readers. English should be improved in this manuscript.

We thank the reviewer for the appreciative comments, recognizing the solidity of our work. We are also grateful for the reviewer’s comment on the English writing. We have carefully and seriously revised the whole manuscript.

Reviewer #3 (Remarks to the Author):

Dear authors,

Your manuscript “Precise control of embolic stroke with magnetized red blood cells” is a smart approach on reversibly induce an embolic stroke. RBCs are loaded with magnetic nanoparticles, such that they react on the magnetic field of a permanent magnet. At the location of the magnet an occlusion can be induced. You show a large variety of different analytical experiments to demonstrate and validate your findings. Please find below some comments and suggestions.

General:

1. Avoid using abbreviation SIMPLE and SIMPLER in the abstract

We apologize for using abbreviations in the abstract. We have written out the full terms in the introduction of the revised manuscript (p.3, lines 79-80 and p. 4, line 84).

2. Please check if you have introduced all the used abbreviations (e.g. MCA?, SMS, CNS, ...) . In my opinion you are using abbreviations very extensively, which makes it sometimes hard to read. Think about not abbreviating everything which you are just using once or twice.

We thank the reviewer for this kind suggestion. We have deleted all the unnecessary abbreviations that were used only once or twice.

3. Just as an idea, which came to my mind: why not using tomographic imaging, such as MRI, which could detect the MNPs as contrast agent? Maybe you could comment or discuss on this?

We used MRI to detect the infarct size and observed the injured brain area (Fig. 3J-K). It would be interesting idea to perform tomographic imaging by using MNPs as a contrast agent. If MNPs

alone are introduced to the bloodstream of healthy mice, we speculate that MRI would detect the MNP signal throughout the brain because the diameters of vessels in the brain are far finer than the spatial resolution of MRI imaging. Moreover, the most abundant vessels in the brain are capillaries with a diameter of 3 to 5 microns, which is much smaller than the spatial resolution of MRI (at several hundred microns). Alternatively, if an optimal dose of mRBCs is introduced to the bloodstream, the result is that only a very small portion of total RBCs in the circulatory system are tagged with MNPs, which might compensate for the limited spatial resolution of MRI. We speculate that if the scanning speed of MRI is twice as fast as the velocity of flowing blood cells, MRI may uncover the movement trajectories of a handful of mRBCs in the brain. Although it is a very interesting topic, we think it is beyond the scope of this article, and we have decided not to include this discussion in the revised manuscript.

Introduction

4. *Fig 2 is introduced before Fig1*

We have the figure numbering in the revised manuscript (p. 4, line 81).

5. You use the word “SIMPLE” before introducing it. The reader just knows that this is the technique done by you before.

We appreciate this feedback. We have introduced the spelled-out form of the abbreviation of SIMPLE in the revised manuscript (p. 4, line 84).

Results

6. *P.4 | 94: “a drop of liquid magnet field gradient”. That is not understandable, you mean something else (which becomes clear when reading the methods). Please rephrase.*

Our apology for the inconvenience caused by English writing. In addition to rephrasing, we have provided more videos (see the revised Movies 1-3) to show how the experiment was performed.

7. P4 I 00: how did you measure the 90s? I suppose the MNPs move and align gradually with the edge of the magnet. When did you stop measuring the time, as soon as you see a line? Or when no MNPs are moving anymore?

Yes, the reviewer is correct. We used the 90th second as the time point to stop measuring, as this was when we saw mRBCs aggregate into a line. The moving speed of mRBCs was slower than that of MNPs. MNPs moved very fast, and no MNPs were moving anymore by the 30th second. Please refer to the revised Movies 1-3.

8. P.6 I 140 (not only here) Why is the magnet you are using a micro-magnet? It is 1mm in diameter (this information should also be mentioned in the methods, I only found it in the figure caption). See my comment about missing information on the used magnets.

We used a cylindrical 1-mm-diameter magnet throughout the study. We apologize for misusing the word 'micro-magnet'. We have changed this term in the revised manuscript, and the related information has been included in the Methods as well (p. 17, line 434-435)

Discussion

9. Dosis limit of MNP: you state "avoid potential longterm toxicity". Can you give a concentration limit when it becomes toxic and what happens then?

Yes, the reviewer is correct. We found that venous administration of an extremely high dose of MNPs, between 150-200 mg/kg, caused considerable lethality in adult mice, demonstrating that this extremely high dose of MNPs is toxic. In the SIMPLE model, although all the mouse pups that received a medium dose of MNPs (80 mg/kg) survived to adulthood, we found MNPs accumulated in the pups' internal organs, such as the liver and spleen, at 6 hours after injection (Fig. S2 and see the revised Movies 10 and 11). In contrast, we did not detect any MNP deposits in the liver or spleen of mouse pups that received the SIMPLER model with a very low MNP dose of 6 mg/kg (Fig. S2 and see the revised Movies 8 and 9). Thus, we anticipated that the deposits in the SIMPLE model might cause long-term side effects on liver function, but we do not have any evidence yet. In our revised manuscript, we have rephrased our statement (p. 8, lines 194-196).

10. Could the agglomeration of uncoated MNPs be a problem, when they are not longer attached to the RBCs?

We thank the reviewer for raising this point. Our results clearly demonstrated that a low dose of MNPs (6 mg/kg) neither reduced blood flow (Fig. 3B and D) nor induced stroke in the SIMPLE model (Fig. 3F and H). This result indicates that even if all the MNPs dissociated from RBCs in the SIMPLER model, the aggregation of uncoated MNPs would not be sufficient to occlude the vessel at all. Thus, we conclude that it would not pose a problem.

11. One drawback of your method is, that only vessels close to surface can be used, because of imaging and because of penetration depth of the magnetic field. Please comment on that and/or discuss possible solution strategies.

Yes, we completely agree. Both the SIMPLE and SIMPLER models have the same drawback: only superficial vessels can be occluded due to the limited penetration depth of the magnetic field. Thus, these models cannot be used to induce deep brain infarction. We have included a comment to this effect in our revised discussion section (p. 13, line 329-331).

We provide some possible solutions below. First, implantation of an electrically controllable magnetic electrode deep in the brain may allow the reversible manipulation of magnetic field induction at the targeted depth. However, one of the pitfalls of this strategy is that the surgery to insert the electrode would cause additional brain injury. In addition, this type of surgery can be performed in adult rodents, but it is very difficult to control the electrode location precisely and reliably in newborn mouse pups. Second, if there were an MRI-like device combining a very strong magnetic field with the ability to control the magnetic field size, one could remotely generate a focal magnetic field gradient in deep brain regions and induce focal ischemia at the desired depth in the brain.

12. Even though small magnet was used, the magnetic field may also affect vessels nearby. Did you observe that? Could it be a problem?

We agree with the reviewer. Although we used a small magnet, it indeed occluded nontarget vessels nearby. This was helpful rather than problematic because the 'distal' middle cerebral artery has numerous collateral vessels that could provide an alternate blood supply around a blocked artery via another path, such as nearby minor vessels. To induce a sufficient reduction in blood flow and hence cause ischemic stroke, it is necessary to block the main artery together with the nearby microvessels.

13. Size and strengths of used magnets are missing

We apologize for omitting this information. We have added information on the size and strength of the magnets in the revised manuscript (p. 17, lines 434-435). Detailed information is shown below as well.

We used one type (1 mm) of magnet that was used in our previous study¹, in which the properties of the magnet were well characterized. The estimated magnetic force of the selected magnet (1 mm) was between 0.25 and 0.5 pN within a distance of 300 μm from the magnet surface, as shown in the graph below, which is reproduced from our previously published paper¹.

14. I could not find all information on every used MNPs. Please try to give comparable information on the used particles

We have now provided clear information on the MNPs in the Methods section (p. 15, lines 386-388, and p. 16, lines 398-399).

In total, we used two types of MNPs, which were from the two sources listed below.

1. The silica-coated magnetic nanoparticles that were used to magnetize RBCs by 'streptavidin-biotin binding were produced by our collaborators at Westlake University (Dr. Botao Ji's laboratory and Dr. Bobo Dang's laboratory).

2. The PEG-2000-coated magnetic nanoparticles that were used for Lipofectamine-mediated RBC magnetization were purchased from MicroMod, a company based in Germany (Cat# 09-82-182 S13218, Nanomag-D, PEG-2000, 180 nm). These custom nanoparticles with a 180-nm diameter were designed specifically for us when we were developing the technology of SIMPLE. The product may not be listed on the supplier's webpage.

We have now mentioned this in the Results section (p. 5, lines 114-115; p. 6, lines 140-141, 155, 161-162).

References

1. Jia, J.M. et al. Control of cerebral ischemia with magnetic nanoparticles. *Nat Methods* 14, 160-166 (2017).

REVIEWERS' COMMENTS:

Reviewer #1 (Remarks to the Author):

I appreciate the authors' revisions, and I have no further comments.

Reviewer #2 (Remarks to the Author):

Accept as it is.

Reviewer #4 (Remarks to the Author):

The authors answered all my raised questions and concerns appropriately. The overall readability and understandability of the revised manuscript and videos are improved. I recommend it for publication in "Nature Communications Biology".